# Novel Quantitative Evaluation of Biotreatment Suitability of Wastewater

**Tianzhi Wang** [1,2,†], **Weijie Wang** [1,†], **Hongying Hu** [2] **and Soon-Thiam Khu** [1,*]

1   School of Environmental Science & Engineering, Tianjin University, Tianjin 300350, China;
    wangtianzhi@tju.edu.cn (T.W.); wangwj_0918@tju.edu.cn (W.W.)
2   Environmental Simulation and Pollution Control State Key Joint Laboratory, School of Environment,
    Tsinghua University, Beijing 100084, China; hyhu@tsinghua.edu.cn
*   Correspondence: soon.thiam.khu@tju.edu.cn
†   These authors contributed equally to this work.

**Abstract:** The development of wastewater treatment industry has gradually entered the high-standard period and the wastewater treatment technology needs to be refined for different types of wastewater. Traditional water quality indicators are not able to explain new problems encountered in the current wastewater treatment process, especially the potential of removing pollutants via biological methods. This research proposed a new method of evaluating the biological treatment process by measuring the oxygen consumption in the biodegradation of pollutants on-the-go and describing the complete biological oxygen consumption process. The biodegradability of wastewater from an actual textile wastewater treatment plant was quantitatively evaluated by analyzing the proportion of different organic pollutions. Results showed that the hydrolytic acidification can improve the biodegradability of textile wastewater by increasing the content of biodegradable organic matter (growth of 86.4%), and air flotation has little effect on the biodegradability of the wastewater. Moreover, the biodegradability of the textile wastewater could be improved by increasing the nitrogen and phosphorus content, which could come from urea and $K_2HPO_4$. Concretely, nitrogen source mainly increases organic matter of rapid bio-treated and organic matter of easy bio-treated by 14.94% and 70.79%, and phosphorus source mainly increases the organic matter of easy bio-treated by 143.75%. We found that the optimum concentration of additional N and P to the textile wastewater was 35 mg/L and 45 mg/L, respectively. This approach holds great application prospects such as risk control, optimizing treatment technology, and management, due to its characteristics of being simple, easy to use, and rapid online implement action.

**Keywords:** wastewater treatment; biodegradability; optimized operating conditions; application; high standard

## 1. Introduction

In recent years, rapid urbanization has brought about the dual dilemma of water quality deterioration and water quantity shortage. In order to utilize reclaimed water as a safe urban "second water source", more sophisticated and innovative wastewater treatment technology must be devised to improve the quality of reclaimed wastewater. As a result, the development of wastewater treatment industry has gradually entered the high-standard period [1]. However, there is a mismatch between traditional water quality indicators, such as chemical oxygen demand (COD), biochemical oxygen demand (BOD), total nitrogen (TN), and total phosphorus (TP), and the new problems in the current wastewater treatment process [2–5]. Therefore, there is a need to explore the relationship between quality characteristics and treatment potential of wastewater in order to develop more efficient water treatment methods [4,6]. Since wastewater contains many types of pollutants with different physical and chemical properties, the treatment (transformation) potential of wastewater may be defined by the difficulty of water quality change during

the process of treatment, storage, distribution, and utilization of wastewater, including water quality stability and treatment characteristics [7,8]. It can be seen that the biological method is often chosen to treat wastewater because bio-treatment of wastewater has the advantage of low cost, relatively simple equipment, providing economic and environmental benefits [9,10]. Therefore, improving the evaluation capabilities of wastewater biological treatment characteristics may be able to optimize wastewater treatment processes.

As we know, wastewater is a complex liquid mixture, containing many types of pollutants with complex physicochemical properties and the concentration of each pollutant varies widely [11]. During the actual treatment process, the total oxygen demand (TOC) and dissolved organic carbon (DOC) are also measured, and wastewater treatment plants often use $BOD_5/COD$, $BOD_5/TOD$, and/or $BOD_5/DOC$ to roughly judge the biodegradability of wastewater (Table 1). However, these comprehensive indicators of organic pollutants only characterize the total amount of pollutants that are easily oxidized and cannot describe the feature and degradation potential of wastewater [12]. In addition, wastewater, especially industrial wastewater, contains not only easily biodegradable substances, but organics that cannot be degraded or even bio-toxic pollutants [13,14]. Taking the textile industry as an example, it consumes a lot of water, discharges a large amount of wastewater, and has great potential for pollution. The common treatment method is to pre-treat in the factory and then discharge it to a municipal wastewater treatment plant for further purification. However, these wastewaters still contain a large number of refractory pollutants, some of which are toxic to microorganisms (more than 1000 μg/L of aniline) or cannot be biodegradable (more than 100 μg/L of antimony) [15], and these pollutants often result in unsatisfactory wastewater treatment effects and substandard effluent water quality [16]. When measuring the treatment characteristics of different dying, coking, and metal processing wastewater, Hu et al. [17] found that the DOC removal rate of wastewater with the same $BOD_5/DOC$ value is very different, for example, the wastewater with $BOD_5/DOC$ was 1.2, and the DOC removal rate changes between 30% and 80% [17]. In addition, the efficiency of conventional biotechnology to remove or transform pollutants from different wastewater treatment plants is also different. Therefore, establishing a systematic and standardized evaluation method for wastewater biological treatment characteristics is an important topic for studying the potential of water quality conversion.

**Table 1.** Simple indicators for evaluation of wastewater biodegradability.

| Indicators | Values | Biodegradability | Citations |
|---|---|---|---|
| $BOD_5/COD$ | 0.4~0.6 | Easily biodegradable | [18] |
| $BOD_5/DOC$ | >1.2 | Easily biodegradable | [17] |
| $BOD_5/COD$ | 0.2~0.4 | Difficult to be biodegradable | [7] |
| $BOD_5/COD$ | <0.1 | Non-biodegradable | [14] |

At present, traditional evaluation methods cannot describe the dynamic process of organics degradation in wastewater [3], and it is difficult to analyze and distinguish the difficulty of biodegradation of organic matter accurately without finding out the conversion mechanism of organics. Therefore, these methods not only fail to guide the application of biological treatment technology correctly, but are unable to guide the combination of biological treatment and other treatment processes [19]. Therefore, it is indispensable to evaluate its biodegradation characteristics comprehensively by determining the components of difficult-to-biodegradable organic matter accurately. How to predict the effect of bio-treatment scientifically and accurately? Is it necessary to join pre- or post-processing when choosing bio-treatment? How to optimize the operating parameters of the bio-treatment process? Research on the above questions is the important content of wastewater quality evaluation, which could provide an important basis for the selection of biological treatment and combined processes.

To address the above challenges, an experimental platform that can directly measure the biological treatment process of wastewater was designed in this study, and the operating

conditions were optimized in this research. The initial time and the difficulty of pollutant degradation can be accurately judged by on-the-go observation of the oxygen consumption, thereby evaluating the characteristics of wastewater biological treatment quantitatively. Finally, we selected the typical textile wastewater with complex organic components as a case to further verify the accuracy of the method for evaluating the wastewater biological treatment characteristics and the effectiveness of optimizing the water treatment process.

## 2. Methodology

### 2.1. Classification of Wastewater Bio-Treatment Characteristics

Biotechnology, with the advantages of low cost and high treatment efficiency (the process can remove 90% of pollutants), has been widely used in the wastewater treatment process [20]. Therefore, understanding and evaluating wastewater bio-treatment characteristics systematically would be an important guideline for selecting and optimizing the appropriate wastewater treatment process. The initial time and the difficulty of pollutant biodegradation could be obtained by monitoring the biological treatment process of wastewater directly and measuring a complete biological oxygen consumption process on-the-go [21]. Then, according to the determination of the oxygen consumption process, different types of wastewaters can be roughly divided into four categories: easy bio-treatment, easy bio-treatment after sludge acclimation, bio-treatment after sludge acclimation, and difficult bio-treatment. The oxygen consumption process and the current situation of treatment in different types of wastewaters are shown in Figure 1. From the perspective of the source of wastewater, the composition of domestic wastewater is relatively simple, and it is most suitable for biological treatment [22]. The biological treatment of aquaculture wastewater and textile wastewater with high concentration of organic matter requires a certain period of biological domestication or adaptation process [23,24].

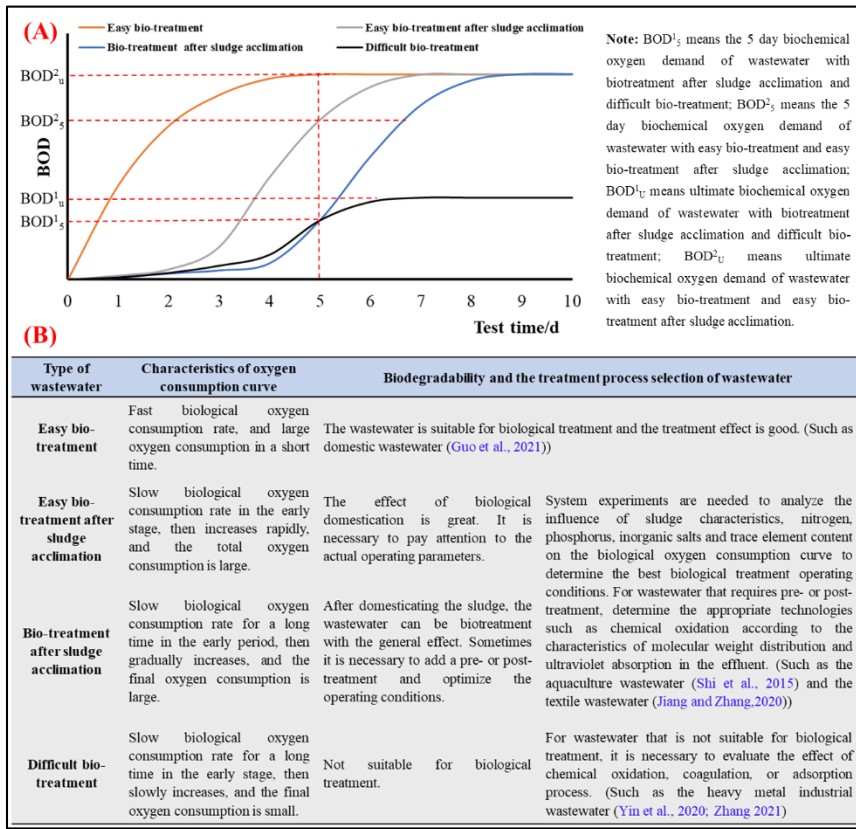

**Figure 1.** Curve diagram of four categories wastewater; (**A**) Oxygen consumption process; (**B**) Classification of biodegradability and current treatment situation.

### 2.2. Bio-Treatment Evaluation Process of Wastewater

This paper proposes a new method for evaluating the biological treatment characteristics of wastewater. Figure 2 is the complete biodegradability evaluation process of wastewater. The following are the specific technical processes:

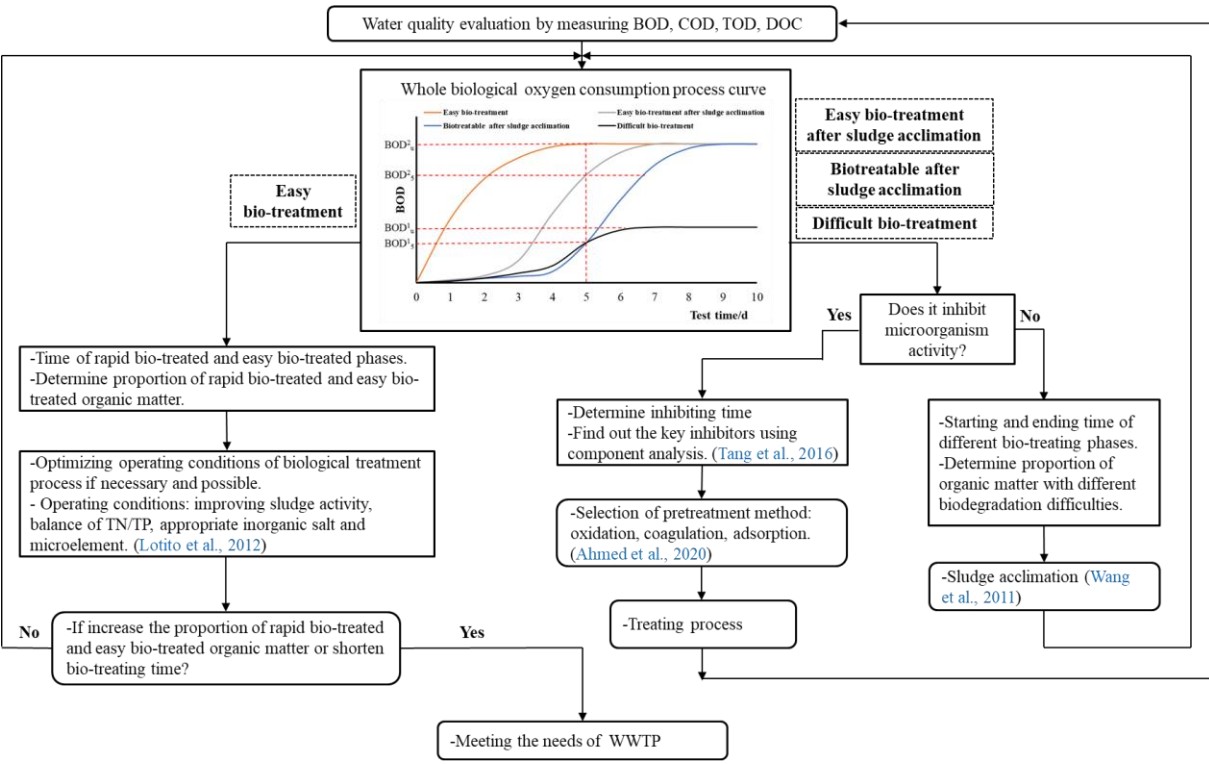

**Figure 2.** A complete biodegradability evaluation process of wastewater.

(1) Determine $BOD_5$, COD, DOC, and TOD. When measuring $BOD_5$, the DOC and COD changes in the water should be measured in order to obtain the removal rate of DOC and COD. The specific measured methods are in Appendix A.

(2) Obtain a complete biological oxygen consumption curve by determining the oxygen consumption during the biodegradation process, and then classify wastewater preliminarily, according to the above four categories and comprehensive indicators of organic pollutants.

(3a) If the wastewater is "Easy bio-treatment", it means that it has excellent biological treatment effects. Then, the oxygen consumption curves can be obtained through experiments. Since the endogenous oxygen consumption process of microorganisms is constant, the different stages of the biological treatment process can be divided according to the changes in the oxygen consumption rate curve [25,26]. At the same time, we can also obtain the ratio and biological treatment time of rapid bio-treated and easy bio-treated organics [15]. The biological treatment efficiency can be improved by improving the sludge activity, maintaining the TN/TP balance, and adding appropriate inorganic salts and trace elements [6,27].

(3b) If the wastewater is "Easy bio-treatment after sludge acclimation", "Biotreatment after sludge acclimation" or "Difficult bio-treatment". It is needed to analyze whether the biological treatment is inhibited according to the rate of oxygen consumption. In addition, we can compare the endogenous respiration process of microorganisms to determine the time when the inhibitory reaction occurs, and then combine the analysis of the pollutant composition to seek out the pivotal inhibitors [7]. Therefore, pretreatment methods such as coagulation, adsorption, and oxidation can be selected to improve the biodegradability of this wastewater [28,29]. If it is not that organisms are inhibited, then we need to determine what organic matter is difficult to biodegrade [30,31]. According to the characteristics of

wastewater quality, wastewater treatment plants usually domesticate sludge to improve the degradation efficiency of pollutants in the sewage [32].

(4) Repeat the above analysis and evaluation procedures to further increase the ratio of rapid bio-treatment and easy bio-treatment organics or shorten the reaction time. The final aim of biotreatment suitability evaluation methodology is meeting the real needs of wastewater treatment plants.

### 2.3. Stage Division of Biological Treatment Process

According to the change characteristics of the oxygen consumption rate curve, the biological treatment process can be divided into three stages: I. Rapid Degradation Stage; II. Constant Degradation Stage; III. Slow Degradation Stage. Furthermore, the organic matter in wastewater can be distinguished as four types: rapid bio-treated, easy bio-treated, normal bio-treated, and difficult bio-treated, which the calculation methods refer to Wang et al. [15].

### 2.4. Bio-Treatment Feature Evaluation Experiment

The experimental test platform for wastewater biological treatment is composed of four parts: aerobic reaction module, $CO_2$ adsorption module, gas detection module, and data processing module. The experimental setup is shown in Figure 3. The process of microbial degradation produces $CO_2$, which is absorbed in the adsorption unit. At the same time, negative pressure is generated in the reaction unit, and $O_2$ is supplemented by the trace gas determination unit (the amount of supplementary gas can be recorded in real time). Among them, the trace gas determination unit has built-in pressure and temperature sensors, and data acquisition is performed based on pulse signals. The device can convert different ambient temperatures and pressures into values under standard conditions, enabling real-time measurement of oxygen consumption. The measured value is transmitted to the data analysis unit through the Internet, and the data analysis and result output processes are completed.

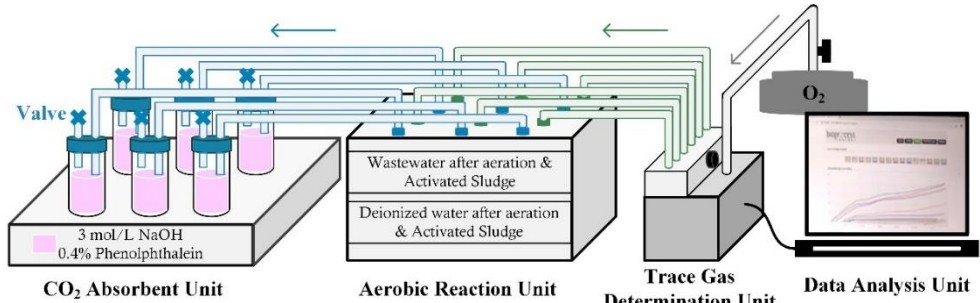

**Figure 3.** Evaluation experiment system of bio-treatment feature.

Sufficient hydraulic retention time and the proper dosage ratio of wastewater and activated sludge can ensure complete contact between microorganisms and pollutants, improve biological treatment efficiency, and allow activated sludge to perform biodegradation [33]. The optimum reaction conditions of the evaluation system have been obtained by analyzing the variation of oxygen consumption using the synthetic wastewater in Appendice B and Appendice C. The mixed liquid of the experimental reaction system is 300 mL, in which the dosage ratio of wastewater and activated sludge are both 50%, and the sludge concentration (MLSS) is 4 g/L. In addition, the biological treatment process is also affected by factors such as the temperature and pH of the reaction system. Therefore, we controlled the temperature and pH to stabilize at 20~25 °C and 7.5 for 10 min, respectively. The stirring rate is 80 rad/min. Under the above conditions, the sludge and organic matter can be fully mixed, and the DO in the reaction flask can be kept at 3–5 mg/L, which is beneficial to the sludge activity. This is consistent with the study by Wu et al. [34], and the reaction

conditions of the bio-treatment feature evaluation experiment we designed are basically consistent with the actual biological treatment process of the textile wastewater plant.

## 3. Application of Quantitative Evaluation for Bio-Treatment in a Wastewater Treatment Plant

### 3.1. Water Quality Analysis of Textile Wastewater

Based on the water quality characteristics of textile wastewater and factory supervision mode, there are two main groups of treating textile wastewater in global: in-plant and secondary treatment. The countries with scattered textile factories would choose the in-plant treatment method, like German [35]. If textile enterprises are centrally constructed and small in scale (the Sewage volume is under 1500 $m^3$/day), secondary treatment is usually chosen, like China and Romania [15,36]. Specifically, the textile wastewater from 6~10 enterprises was preprocessed in factories and then transported to the nearest WWTP together. The concentration of COD is about 10,000 mg/L in textile wastewater, and it will still contain over 300 mg/L COD even using inhouse treatment plants [20]. It cannot be ignored that textile wastewater also contains a variety of dyes, some of which are even toxic to microorganisms, resulting in it being difficult to simply degrade by biodegradation. Therefore, it is necessary to judge the characteristics of textile wastewater quantitatively by proposing a new way to evaluate the biological treatment process of wastewater. As mentioned above, we chose a textile wastewater treatment plant in Jiangsu Province, China as a research case to verify the effectiveness and practicability of the evaluation method in this paper due to most of the textile enterprises (around 1500) being concentrated in Jiangsu provinces. The textile wastewater, which was collected from Shengze Wastewater Treatment Plant in Wujiang District, Suzhou City, China (30°53′34″ N, 120°38′16″ E), was chosen for application of quantitative evaluation for bio-treatment. This wastewater treatment plant accepts industrial wastewater from 8 surrounding textile plants, with a daily treatment capacity of 7000 $m^3$, and the final effluent enters Taihu Lake. The existing treatment process combination of the textile wastewater treatment plant [37] is shown in Figure 4. Textile wastewater has a complex and changeable composition, and the direct biochemical treatment is prone to inhibit the activity of sludge. The textile wastewater usually first performs air flotation treatment after the conditioning tank to remove the emulsified oil that is difficult to settle or the tiny suspended particulate matter. Then, the hydrolytic acidification process is used to convert the macromolecular substances into easily biodegradable small molecular substances, thereby improving the biodegradability of wastewater and providing a good water quality environment for subsequent biochemical treatment. After the biodegradation of Anaerobic-Anoxic-Oxic (AAO) process and precipitation, the wastewater treatment plant added a biological activated carbon tank to further degrade the organic matter. Finally, the effluent that meets the water quality standard will be discharged into the river after ultraviolet (UV) disinfection (Figure 4).

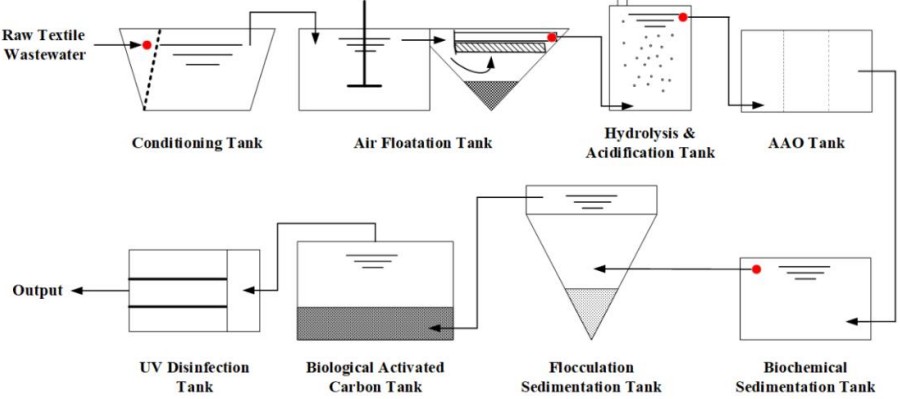

Note: AAO means Anaerobic-Anoxic-Oxic; UV means ultraviolet; the red dots represent the sampling locations.

**Figure 4.** Existing textile wastewater treatment process.

By collecting and testing the water samples at four locations of the treatment plant at the front of the conditioning tank, after the air flotation tank, after the hydrolysis and acidification tank, and after the biological treatment process (Figure 4), we can basically grasp the changes in water quality throughout the water purification process. Among them, air flotation and hydrolytic acidification are commonly used pretreatment processes, which respectively improve the biodegradability of wastewater by physical impurity separation and extracellular biochemical decomposition of macromolecular organic matter [15,37]. The pollutant concentrations of textile wastewater in different treatment stages are shown in Table 2. The B/C of the influent is 0.20, indicating its low biodegradability [38]. Hydrolytic acidification technology can enhance the high concentrated organic wastewater significantly (increased by 55%), with a B/C value of 0.31. However, the air flotation technology has little effect on the biodegradability of the wastewater. After the biological treatment process, the COD of the textile wastewater is 153 mg/L, and the removal rate can reach 37.3%, which is consistent with its biodegradability results. The main function of air flotation is to remove SS, and the removal rate can reach 63.3%, but the substances contained in SS are mainly flocs and inorganic particles [39], so that the removal rate of COD by air flotation is only 7.1%. Hydrolysis acidification technology can degrade macromolecular organic matter into small molecular organic matter [40], and the removal rate of COD is 18.7%. In addition, the BOD content in the wastewater increased by 27.9%, which is conducive to the next stage of microbial degradation process. However, the absolute content change of biologically treatable organics in wastewater cannot be merely obtained by the B/C.

**Table 2.** Concentrations of pollutants in textile wastewater at different treatment stages.

| Water Quality | COD (mg/L) | BOD$_5$ (mg/L) | SS (mg/L) | TN (mg/L) | TP (mg/L) | B/C | C:N:P |
|---|---|---|---|---|---|---|---|
| Raw textile wastewater | 323 | 64 | 215 | 12.8 | 0.33 | 0.20 | 979:39:1 |
| Textile wastewater after air floatation | 300 | 61 | 79 | 11.6 | 0.31 | 0.20 | 968:37:1 |
| Textile wastewater after hydrolysis acidification | 244 | 78 | 63 | 10.5 | 0.25 | 0.32 | 976:45:1 |
| Textile wastewater after AAO process | 153 | 15 | 32 | 3.5 | 0.17 | 0.10 | 423:21:1 |

It can be seen from Table 2 that the C/N/P of textile wastewater is generally much higher than 100:5:1, indicating that the lack of nitrogen and phosphorus is not conducive to biological treatment. This conclusion stays in step with the study of Wang et al. [15]. The case study also shows that the COD in the secondary effluent of the existing textile wastewater treatment process is 153 mg/L, which is higher than the requirement (100 mg/L) in the "Water Pollutant Discharge Standard for Textile Dyeing Industry" (GB 4287-2012) and the emission limit of 60 mg/L required by the first level B standard of the "Pollutant Discharge Standard for Urban Wastewater Treatment Plants" (GB 18918-2002), which cannot be directly discharged into natural receiving water bodies. The current treatment method adopted by the wastewater treatment plant is to add a sedimentation tank after the secondary effluent to ensure the separation of mud and water, and then use a biological activated carbon filter to further adsorb and remove the organic matter in the wastewater, and finally reduce the COD concentration to about 30 mg/L [41]. It is worth noting that a series of processes added by the water plant after AAO are all because the pretreatment process fails to adjust the biodegradability of wastewater to the best state, which undoubtedly increases the risk of uncertainty for the effluent quality [42]. Therefore, the effect of each stage of the water treatment process, especially the pretreatment stage, needs to be evaluated. Based on this, the wastewater treatment plant can improve the operating conditions in a targeted manner, thereby reducing costs and improving the stability of effluent quality.

*3.2. Effect of Existing Treatment Processes on the Wastewater Biodegradability*

The oxygen consumption rate and cumulative oxygen consumption of textile wastewater in different biological treatment stages can be determined through the experiment

platform. According to the oxygen consumption rate curve in Figure 5a, the organic matter in wastewater can be divided into the four types introduced in Section 2.3.

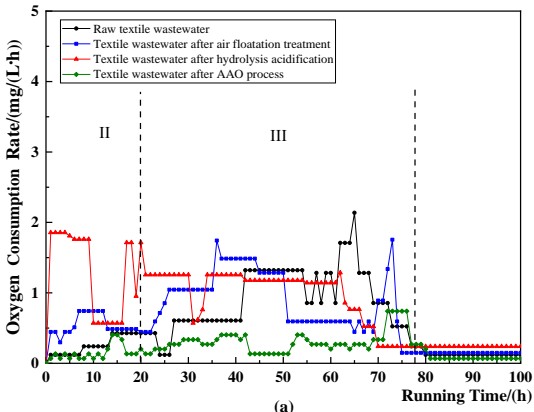
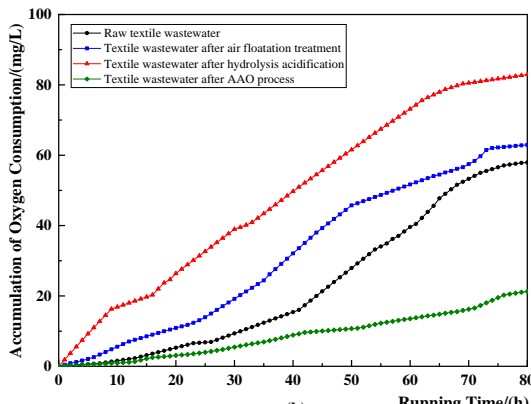

**Figure 5.** The oxygen consumption process of textile wastewater in different treatment stages ((**a**) Oxygen consumption rate; (**b**) Cumulative oxygen consumption).

According to the cumulative oxygen consumption curve in Figure 5b, the distribution of the difficulty in degradation of organic matter in sewage is obtained (Table 3). The textile wastewater that has not been treated by air flotation and hydrolysis acidification cannot be rapidly degraded by activated sludge. Combining the ratio of BOD and COD concentration, it can further illustrate that the biodegradability of textile wastewater is poor. This is because the wastewater contains many difficult-to-degrade organics such as synthetic slurries, dyes, and surfactants with complex molecular structures [15]. Dyes containing polar auxochromic groups are difficult to decolorize and mineralize, and it may need advanced oxidation processes to remove them [43]. High-salt dyeing auxiliaries will affect cell osmotic pressure and inhibit microbial growth [44]. In addition, the higher the solubility of the dye, the more difficult it is to decolorize wastewater. Materials with higher adsorption efficiency are required to participate in the treatment process [45]. The colloids in textile wastewater are destabilized under the action of air flotation technology, forming small granules, which then gather into larger granules and deposit, removing the superfluous pollutants [5]. The air flotation treatment can remove most of the SS and part of COD in wastewater, increasing the biodegradable organic matter (from 17.6% to 20.7%).

**Table 3.** Degradability of organic matter (calculated as COD) in wastewater among treatment process.

| Type of Organic Pollutions | Rapid Bio-Treated | Easy Bio-Treated | Normal Bio-Treated | Non-Biological Treatment |
|---|---|---|---|---|
| Raw textile wastewater | 0 | 5 | 52 | 266 |
| Textile wastewater after air floatation treatment | 2 | 7 | 53 | 238 |
| Textile wastewater after hydrolysis acidification | 5 | 24 | 51 | 164 |
| Textile wastewater after AAO process | 0 | 3 | 17 | 133 |

In addition, after being hydrolyzed and acidified, the biodegradability of textile wastewater has been significantly improved. The content of biodegradable organic matter increased from 17.6% to 32.8% (growth of 86.4%), of which the rapid bio-treated organic matter increased from 2 mg/L to 5 mg/L, the content of easy bio-treated organic matter increased by more than 3 times, from 7 mg/L to 24 mg/L. This is because in the process of hydrolysis and acidification, some facultative anaerobic bacteria (such as *Clostridium*, Anaerobic *peptococcus*, *Escherichia coli*, etc.) decompose the difficult-to-degrade high molecular polymers and heterocyclic organics in the wastewater into easy-to-degrade organic alcohol or acid with small molecule [46,47]. The increase of these small organic molecules

greatly promoted the respiration rate of microorganisms, which increased the respiration rate from 0.32 mg/(L·h) to 1.86 mg/(L·h).

### 3.3. Effect of Extra Nutrition on Bio-Treatment of Textile Wastewater

According to the results in Section 3.2, the hydrolysis acidification cannot improve the biodegradability of wastewater obviously, and there are only 32.8% of the pollutants that can be degraded by microorganisms, and the total degradation time is up to 80 h. Because textile wastewater is deficient in nitrogen and phosphorus, which is not conducive to biological treatment, we add urea with concentration gradients of 20, 35, and 50 mg/L as the supplement nitrogen source, and added $K_2HPO_4$ with concentration gradients of 15, 30, and 45 mg/L as the supplement phosphorus. The addition of different types of nutrient sources are shown in Table 4. Nitrogen and phosphorus are typically chosen for improving biological treatment characteristics, and traditional research focuses on the effect of nitrogen and phosphorus on the microorganism of activated sludge [27,47]. However, traditional evaluation methods cannot describe the dynamic process of organic degradation in wastewater under different external nutrients [3]. It is indispensable to evaluate the effect of nitrogen and phosphorus on biodegradation characteristics comprehensively by determining the organic matter degradation difficulty accurately.

**Table 4.** Types of nutrient sources of test.

| Nutrient Sources | Dosing Concentration (mg/L) | Concentration after Input into Textile Wastewater (mg/L) | |
| --- | --- | --- | --- |
| | | **TN** | **TP** |
| Urea | 20 | 16.54 | — [a] |
| | 35 | 23.88 | — [a] |
| | 50 | 30.35 | — [a] |
| $K_2HPO_4$ | 15 | — [a] | 2.72 |
| | 30 | — [a] | 6.04 |
| | 45 | — [a] | 7.89 |

[a] No this type of additional nutrient sources was added to the wastewater.

The rate and the accumulation of oxygen consumption change characteristics of the biological treatment process under different external nutrient sources are shown in Figure 6, and the distribution results of the organic matter degradation difficulty are shown in Table 5. It can be seen from Table 5 that the supplementation of nitrogen and phosphorus sources significantly affects the biological treatment process, especially improving the treatment effect in the rapid and easy bio-treated organic matter. As the nutrient concentration increases, the content of organic matter that can be biologically treated in the textile wastewater gradually increases, which can be increased from 18.13% to 29.06%, and the increase rate can reach 53.45–60.34%.

In the case of using nitrogen source alone, the average oxygen consumption rate of the 35 mg/L concentration group is the highest at the stage I, which is 2.68 mg/(L·h), 33.18% higher than the 45 mg/L concentration group. The blank group and the 20 mg/L concentration group do not have a rapid degradation stage. After 5 h of reaction, the oxygen consumption rate of all the treatment groups with the addition of nitrogen source further increased. The oxygen consumption rate of the 35 mg/L concentration group reached its peak at the 16 h, which was 3.42 mg/(L·h), 92.31%, 4.17%, and 5.63% higher than the peak oxygen consumption rate of the blank group, 20, and 50 mg/L concentration group, respectively. After 57 h of reaction, the oxygen consumption rate of wastewater stabilized at 0.33~0.86 mg/(L·h), which means the microorganisms entered the endogenous respiration stage and no longer degraded organic matter. In addition, it can be seen from Table 5 that compared with the blank group, the use of nitrogen sources increased the content of rapid and easy bio-treated organics. Among them, the 35 mg/L concentration group had the highest proportion of rapid bio-treated organics, which was 14.94%, which was 30% higher

than the 50 mg/L concentration group. The 50 mg/L concentration group had the highest proportion of easy bio-treated organics, which was 70.79%, which are 36.13% and 8.04% higher than the 20 mg/L and 35 mg/L concentration groups, respectively.

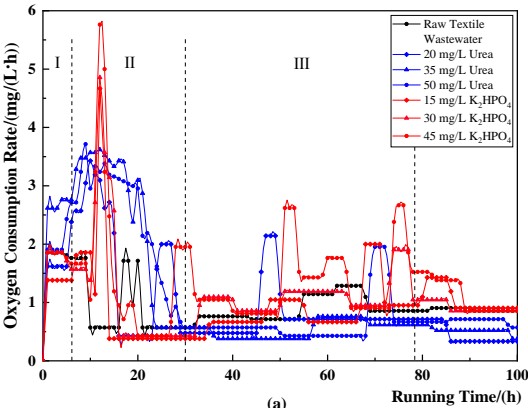
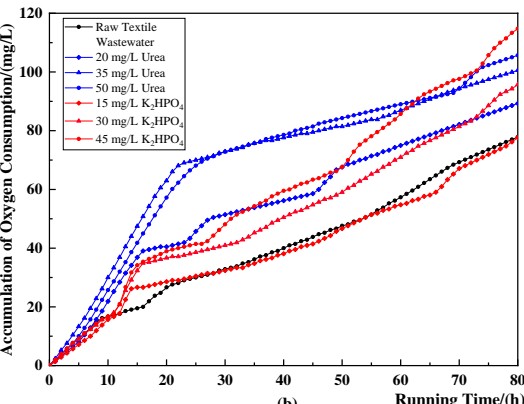

**Figure 6.** Oxygen consumption process curve of textile wastewater under different nutrient sources ((**a**) Oxygen consumption rate; (**b**) Cumulative oxygen consumption).

**Table 5.** Degradability of organic matter (calculated as COD) after adding different nutrients.

| Type of Organic Pollutions | | Rapid Bio-Treated (mg/L) | Easy Bio-Treated (mg/L) | Normal Bio-Treated (mg/L) | Non-Biological Treatment (mg/L) |
|---|---|---|---|---|---|
| KB [a] | | 0 | 16 | 42 | 262 |
| Supplement dosage of nitrogen (Urea) | 15 | 0 | 39 | 36 | 245 |
| | 30 | 13 | 57 | 17 | 233 |
| | 45 | 10 | 63 | 16 | 231 |
| Supplement dosage of phosphorus ($K_2HPO_4$) | 14 | 0 | 26 | 25 | 269 |
| | 28 | 0 | 35 | 42 | 243 |
| | 42 | 0 | 39 | 54 | 227 |

[a] The blank group, that is, no nutrient source has been added.

In the case of using a phosphorus source alone, the oxygen consumption rate of the microorganisms is maintained at about 1.71 mg/(L·h) during 0–10 h, which indicates that the microorganisms are in the endogenous respiration stage at this time and are adapting to the phosphorus source. After 10 h, the oxygen consumption rate of all phosphorus source treatment groups increased rapidly, reaching a peak at the 12 h. Among them, the 45 mg/L concentration group had the largest oxygen consumption rate, which was 5.76 mg/(L·h), 23.47%, and 18.63% higher than the 15 and 30 mg/L concentration groups, respectively. After 55 h, the oxygen consumption rate of wastewater stabilized at 0.67–1.90 mg/(L·h), and the microorganisms entered the endogenous respiration stage and no longer degraded organic matter. In addition, it can be seen from Table 5 that compared with the blank group, the use of phosphorus sources increased the content of easy bio-treated organics. The 45 mg/L concentration group had the largest increase, with an increase of 143.75%, which was 50.00% and 11.43% higher than the 15 and 30 mg/L concentration group, respectively.

The above results indicate that the textile wastewater lacks sufficient nitrogen and phosphorus sources to react in biological treatment process. The ideal C:N:P ratio required by microorganisms is 100:5:1 [48], but the actual ratio of the textile wastewater is 979:39:1, so it should be supplemented by nitrogen and phosphorus sources to improve the microbial activity, and then improve the biological treatment effect, which can be proved by the experimental results. Meanwhile, in this study, the optimum concentration of urea (nitrogen

source) and $K_2HPO_4$ (phosphorus source) added to textile wastewater was 35 mg/L and 45 mg/L.

The final concentration of TN and TP also had been measured and the results of continuous monitoring for 38 days are shown in Figure 7. The concentrations of TN and TP were lower than 14 mg/L and 0.5 mg/L, respectively, which both met the discharge standards of wastewater treatment. It shows that adding nitrogen and phosphorus nutrient sources to the wastewater will not have a negative impact on the final effluent quality. The research results of supplementing nitrogen and phosphorus to optimize the efficiency of biological treatment have been applied to the Shengze Wastewater Treatment Plant.

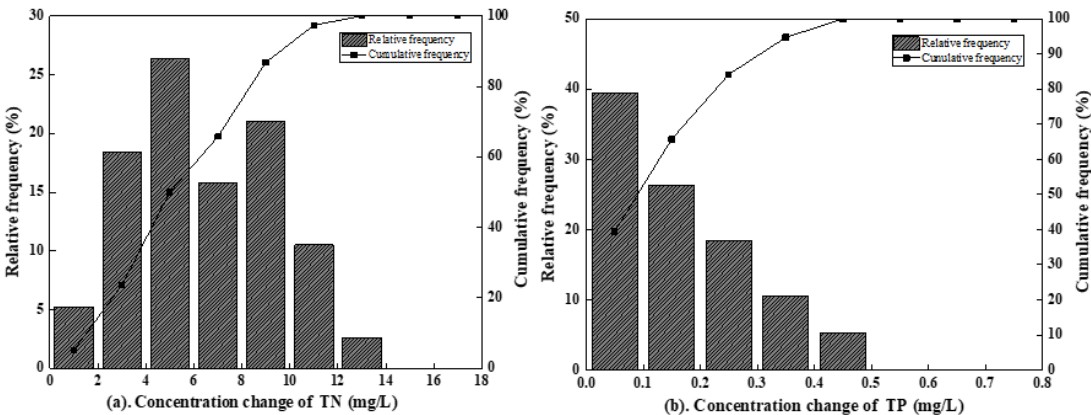

**Figure 7.** The final concentration of TN and TP ((**a**) Concentration change of TN; (**b**) Concentration change of TP).

## 4. Conclusions

In order to make reclaimed water a safe urban "second water source", wastewater treatment has gradually entered the high-standard stage and the wastewater treatment technology need to be refined for many kinds of wastewater. The source of wastewater is complex and changeable, resulting in constant new problems that plague the water purification process of sewage treatment plants, which seriously affects the quality of effluent water. Firstly, the component characteristics of the pollutants themselves have an impact on the microbial activity of sludge. Secondly, the organic pollutants will be transformed into each other, and the biological treatment process of different pollutants may have cooperation or competition due to the transformation of raw materials and products. The traditional water quality indicators not only fail to guide the application of biological treatment technology correctly, but are unable to guide the combination of biological treatment and other treatment processes. To meet the above challenges, the evaluation of the biological treatment characteristics of wastewater needs to be quantified and accurate. This research proposed a new biological treatment characteristic evaluation method by using the oxygen consumption measurement results of pollutants in the biodegradation process to find out the complete biological oxygen consumption process, the starting and the consuming time of the biological treatment equipment, and then analyzing the proportion of different biodegradable organics in wastewater. The optimum reaction conditions of the evaluation system have been obtained by analyzing the variation of oxygen consumption using the synthetic wastewater. This evaluation method not only improves the accuracy of the evaluation results but also remedies a defect of the traditional method that cannot be quantified by $BOD_5/COD$, $BOD_5/TOD$, and $BOD_5/DOC$. Therefore, the quantitative evaluation for bio-treatment of wastewater can be used as a step in the evaluation of water characteristics to help relevant staff or scholars to deepen their understanding of wastewater treatment processes and water quality standards, adjust operating parameters, optimize existing processes, or develop new processes.

We took a real wastewater treatment plant in Suzhou, Jiangsu Province as a research case to evaluate the biodegradability of wastewater further quantitatively and improv-

ing methods. On the one hand, the hydrolytic acidification technology can increase the biodegradability of textile wastewater by increasing the proportion of easy bio-treated organics, and air flotation has little effect on the biodegradability of the wastewater. On the other hand, the condition of nitrogen and phosphorus in wastewater is very unfavorable for biological treatment. The content of biodegradable organic matter could be increased by increasing the nitrogen source and phosphorus source in the textile wastewater. The optimum concentration of urea (nitrogen source) and $K_2HPO_4$ (phosphorus source) added to the textile wastewater was 35 mg/L and 45 mg/L, respectively. Nitrogen source mainly increases the proportion of rapid and easy bio-treated organics by 14.94% and 70.79%, and phosphorus source mainly increases organic matter of easy bio-treated by 143.75%. The result of this research has been put into practice in Shengze Wastewater Treatment Plant for optimizing the reaction time and the supplement dosage of nitrogen or phosphorus in the biological treatment process.

**Author Contributions:** Conceptualization, T.W. and H.H.; methodology, T.W. and W.W.; validation, S.-T.K. and H.H.; data curation, T.W. and W.W.; writing—original draft preparation, T.W. and W.W.; writing—review and editing, T.W. and W.W.; supervision, S.-T.K.; funding acquisition, S.-T.K. and H.H. All authors have read and agreed to the published version of the manuscript.

**Funding:** This research was funded by National Natural Science Fund of China [No. 51321001, 51808314] and the Independent Innovation Fund of Tianjin University [2021XZ-0019].

**Acknowledgments:** Thanks to the editors and reviewers for their valuable comments and suggestions for this article.

**Conflicts of Interest:** There is no conflict of interest in this paper.

## Appendix A

**Table A1.** Measured Methods of Common Biochemical Indicators.

| Pollutant Indicators | Measured Methods | Standard/Instrument |
| --- | --- | --- |
| COD | Dichromate oxidation | ISO6060 |
| BOD | Differential pressure detection | Automatic BOD Tester |
| TOC | Combustion | TOC Analyzer |
| SS | Weigh after filtration | 0.45-micron filter, balance |
| TN | Potassium Persulfate Oxidation, Spectrophotometry | Spectrophotometer |
| TP | Ammonium molybdate, spectrophotometry | |

## Appendix B

When microorganisms in activated sludge degrade organic matter in wastewater, sufficient hydraulic retention time is required to ensure the complete contact between microorganisms and pollutants to improve biological treatment efficiency. In addition, the biological treatment process is also affected by factors such as the temperature and pH of the reaction system. In this test system, mechanical agitation is selected to increase the mixing degree of wastewater and activated sludge. In the optimization experiment, artificially synthesized wastewater is used, and the formula is as shown in Table A2. Among them, COD = 510 ± 12 mg/L, BOD = 387 ± 17 mg/L. The mixed liquid of the experimental reaction system is 300 mL, in which the dosage ratio of wastewater and activated sludge are both 50%, the sludge concentration (MLSS) is 4g/L, and the other operating conditions are shown in Table A3.

The activated sludge suspension was collected from the outlet of the aeration tank of the biological treatment unit, and then it was cleaned in the laboratory immediately. The cleaning steps were the following: (1) After centrifuging the suspension (4000 r/min, 10 min), the supernatant was discarded following a clear muddy water boundary appearing. (2) The bottom sludge was resuspended three times with tap water, which had been dechlorinated by aerating with air. After cleaning, a small part of sludge was dried in

the equipment at 105 °C, and then its dry weight was measured to get the mixed liquid suspended solids (MLSS). Most of the activated sludge suspension was stored at 4 °C for refrigeration, and the storage time should not exceed 3 days. Before the experiment, the researcher adjusted the MLSS of the refrigerated activated sludge to an inoculum with the required MLSS.

**Table A2.** Composition and concentration of synthetic wastewater.

| Ingredient | Concentration (mg/L) | Ingredient | Concentration (mg/L) |
|---|---|---|---|
| Glucose | 500 | $CaCl_2$ | 60 |
| Starch | 400 | $(NH_4)_2SO_4$ | 250 |
| Petone | 200 | $K_2HPO_4$ | 60 |
| Urea | 60 | $MnSO_4$ | 8 |
| $NaHCO_3$ | 300 | $FeSO_4$ | 1.2 |
| NaCl | 400 | | |

**Table A3.** The reaction condition parameters of experimental system.

| Reaction Condition | Parameter |
|---|---|
| Stirring rate | 10, 40, 80, 100 rad/min |
| Temperature | 10 °C, 15 °C, 20 °C, 25 °C, 30 °C |
| pH | 5, 7, 9 |

## Appendix C. System Optimization Results of Evaluation for Wastewater Bio-Treatment

### Appendix C.1. Effect of Temperature on the Bio-Treatment Process

The oxygen consumption process of wastewater under different temperature conditions is shown in Figure A1, and as a whole, there is a three-stage change from a slow increase to a rapid increase to a slow linear increase. From 0 to 8 h, the cumulative oxygen consumption increases slowly, reaching 17–59 mg/L, indicating that microorganisms are adapting to the wastewater environment in this stage. From 8 to 28 h, it is in a stage of rapid growth, and the cumulative oxygen consumption can reach 63–214 mg/L, indicating that microorganisms are accelerating the degradation of organic matter in wastewater. After 28 h, the oxygen consumption shows a slow linear growth trend, at this time most of the organic matter in wastewater has been degraded, and microorganisms are in an endogenous respiration stage.

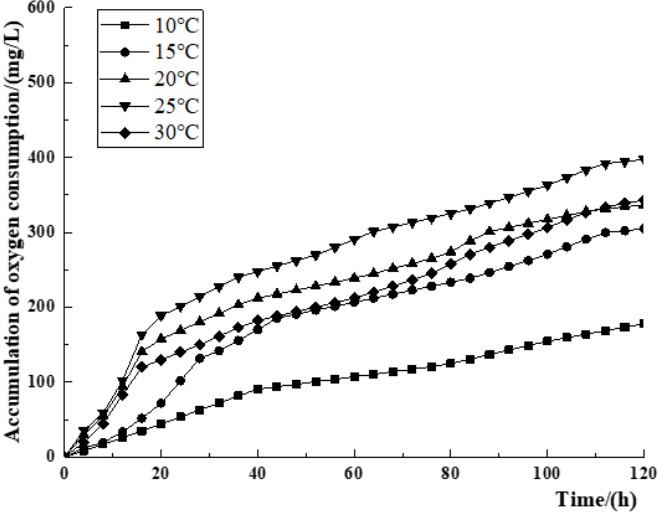

**Figure A1.** Accumulation of oxygen consumption under different temperature conditions.

Peculiarly, in the stage of organic matter rapid degradation, when the reaction temperature increases from 10 °C to 25 °C, the oxygen consumption increases, and when the temperature is higher than 25 °C, the oxygen consumption decreases. At 28 h, the cumulative oxygen consumption in 25 °C was the largest (214 mg/L), which was 239.68%, 62.33%, 18.56%, and 42.51% higher than that under 10 °C, 15 °C, 20 °C, and 30 °C, respectively.

*Appendix C.2. Effect of pH on the Biological Treatment Process*

The oxygen consumption process of wastewater under different pH is shown in Figure A2, and as a whole, there is a three-stage change from a slow increase to a rapid increase to a slow linear increase. From 0 to 4 h, the cumulative oxygen consumption increases slowly, reaching 15–34 mg/L, indicating that microorganisms are adapting to the wastewater environment in this stage. From 4 to 40 h, it is in a stage of rapid growth, and the cumulative oxygen consumption can reach 205–355 mg/L in the end of the stage, indicating that microorganisms are accelerating the degradation of organic matter in wastewater. After 40 h, the oxygen consumption shows a slow linear growth trend, at this time most of the organic matter in wastewater has been degraded, and microorganisms are in the endogenous respiration stage.

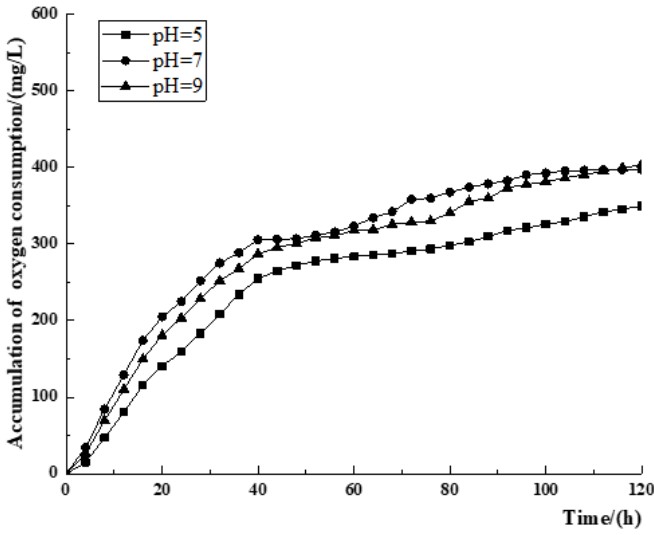

**Figure A2.** Accumulation of oxygen consumption under different pH conditions.

In the stage of organic matter rapid degradation, when the pH increases from 5 to 7, the oxygen consumption increases, and when the pH increases from 7 to 9, the oxygen consumption decreases. At 40 h, the cumulative oxygen consumption reaches the largest (355 mg/L) at pH = 7, which was 19.89% and 6.47% higher than that under pH = 5 and pH = 9 respectively.

Therefore, the optimum reaction conditions of the evaluation system have been obtained by analyzing the variation of oxygen consumption using the synthetic wastewater (25 °C, pH = 7, stirring rate with 80 rad/min).

*Appendix C.3. Effect of Stirring Rate on the Biological Treatment Process*

The oxygen consumption process of wastewater under different stirring rate conditions is shown in Figure A3, and as a whole, there is a three-stage change from a slow increase to a rapid increase to a slow linear increase. From 0 to 4 h, the cumulative oxygen consumption increases slowly, reaching 5–29 mg/L, indicating that microorganisms are adapting to the wastewater environment in this stage. From 4 to 36 h, it is in a stage of rapid growth, and the cumulative oxygen consumption can reach 277–367 mg/L in the end of the stage, indicating that microorganisms are accelerating the degradation of organic matter in wastewater. After 36 h, the oxygen consumption shows a slow linear growth trend, at this time most of the

organic matter in wastewater has been degraded, and microorganisms are in endogenous respiration stage.

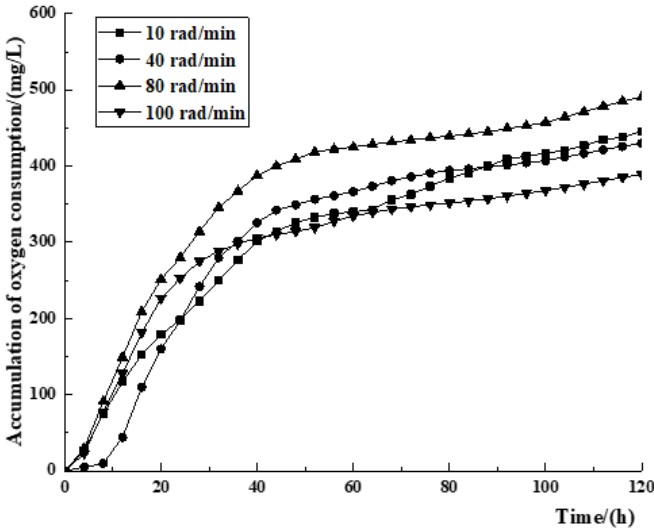

**Figure A3.** Accumulation of oxygen consumption under different stirring rate conditions.

In the stage of organic matter rapid degradation, when the stirring rate increases from 10 rad/min to 80 rad/min, the oxygen consumption increases, and when the stirring rate is higher than 80 rad/min, the oxygen consumption decreases. At the stirring rate of 80 rad/min condition, the cumulative oxygen consumption is the largest (367 mg/L), which was 32.47%, 21.80%, and 23.26% higher than at the 0 rad/min, 40 rad/min, and 100 rad/min, respectively.

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
