# Peer review of "Novel Quantitative Evaluation of Biotreatment Suitability of Wastewater"

_water, doi:10.3390/w14071038_

Round 1
Reviewer 1 Report
Identifying and implementing efficient methods of effluents loaded with hardly biodegradable organic compounds from the textile industry is a research direction of major practical importance. The novelty and originality of the proposed method are moderate.
The introduction partially justifies the need and purpose of this experimental study. Research methodology is structured in distinct subchapters, but a large part of this section is still addressed to the study of bibliographic references although this chapter should more clearly present the methodology used in this study. I recommend a more concise organization and presentation of the methods and equipment used in this research, a clearer presentation of the procedure proposed for the quantitative evaluation. Figures 1 and 2 must be redrawn, the text cannot be clearly distinguished. The results of the study are presented accordingly; they are discussed using arguments based on the investigated articles.
There are some typos, I recommend paying more attention to expressing ideas. Some examples of typos: Line 301: <organic matter increased from 2 mg / L increasing to 5 mg / L>; Line 303: the name of the bacteria must be written with Italics; Lines 402-404: the sentence has no verb; it should be reformulated.
Bibliographic references are relevant for the proposed topic; quite a few are relatively recent.
Author Response
- We have described our methods in “Methodology”, including the equipment and the mechanism for the quantitative evaluation of bioprocessing properties.
- We have improved the clarity of Figures 1 and 2.
- Thanks to the reviewer's reminder, we have re-read and checked the full text, making corrections for some spelling errors and grammatical issues.
Reviewer 2 Report
The work «Novel Quantitative Evaluation of Biotreatment Suitability of Wastewater» is very interesting and actual, because ecological problems, especially about water resources, are essential for a favorable life of the population. New approaches to improve the quality of wastewater treatment and increase control over the degree of treatment is an important area of scientific research.
The question is, what are the prospects for using the proposed approach, given that biological treatment is not widely used and is not considered the most effective?
How appropriate will this approach be for a combined wastewater treatment method?
Author Response
- At present, biological treatment is the most economical and popular advance treatment process in municipal wastewater treatment plants. However, biological treatment processes to only remove easily degradable BOD, and harmful and persistent substances might reduce the efficiency of biological treatment processes. To address those challenges, it is common to treat textile wastewater by using advanced technologies such as coagulation, oxidation, membrane separation, adsorption, and combination process. However, it is difficult to determine which treatment method is most effective based on only water quality index values. Hence, it is necessary to develop a quick method of biodegradability assessment based on evaluating the characteristics of wastewater.
- This approach in our paper holds a great promise for applications such as investigate wastewater feature deeply, treatment technology optimization, risk control and management due to its characteristics of simple, easy to use, and rapidly online implement.
Reviewer 3 Report
1- The introduction is long, make it more precise
2- enhance the resolution of figure 1 and 2
3- how does the Extracellular polymeric substances (EPS) change upon the addition of N and P
4- add a Schematic representation of the bioreactor
5- how does the pollutants concentration affect the biotreatment process (deactivate the micro-organism population)
6- does the solubility and polarity of the dyes in textile wastewater affect its treatment process
Author Response
- We have subtracted the repetition of the introduction to make it more concise.
- We have improved the clarity of Figures 1 and 2.
- In this paper, we mainly focus on the changes in organic matter degradation efficiency after adding N and P to the textile wastewater, in order to prove that our evaluation method is effective and applicable. Extracellular polymeric substances are important components of activated sludge flocs, including proteins, polysaccharides, nucleic acids, etc. The addition of N and P promotes the formation of these substances. The organic matter in the wastewater can be degraded into small molecules by extracellular polymeric substances and absorbed into the microbial cells, thereby improving the biological treatment efficiency of wastewater.
- The main structure of the aerobic reaction unit in the evaluation experiment system of bio-treatment feature evaluation (Figure 3) in this paper is the transport interface for incoming and outgoing gases. The biological reaction process is completely determined by the tested wastewater components, and there is no artificially constructed bioreactor.
- The refractory organics in wastewater are often complex in composition, which may affect the biological treatment process by two ways. First, the component characteristics of the pollutants themselves have an impact on the microbial activity of sludge. Second, the organic pollutants will be transformed into each other, and the biological treatment process of different pollutants may have cooperation or competition due to the transformation of raw materials and products. Taking aniline in textile wastewater as an example, studies have shown that aniline has a strong inhibitory effect on autotrophic nitrifying bacteria. When its concentration is 1 mg/L, the inhibition rate of ammonia oxidation has reached 50% [1]. In addition, aniline has certain toxic effects on activated sludge and nitrifying bacteria [2,3].
- Thanks for your question. There are many kinds of textile fibers. In order to dye different colors, it is necessary to use dyes of different types and structures, and also need to add some dyeing auxiliaries [4]. Dyes containing polar auxochromic groups are difficult to decolorize and mineralize, and it may need advanced oxidation processes to be removed them [5]. High-salt dyeing auxiliaries will affect cell osmotic pressure and inhibit microbial growth [6]. In addition, the higher the solubility of the dye, the more difficult it is to decolorize wastewater. Materials with higher adsorption efficiency are required to participate in the treatment process [7]. We believe that the evaluation method of biological treatment characteristics proposed in this paper can be used to provide a theoretical basis for the treatment process optimization of textile wastewater containing different dyes.
Please see the attachment for the references.

Reviewer 4 Report
The article is written in a very detailed and methodical way. Research planned and carried out correctly. Applications appropriate. Minor correction required Fig. 1 and Fig. 2 are difficult to read.
Author Response
Thanks for your suggestion. We have improved the clarity of Figures 1 and 2.
Round 2
Reviewer 3 Report
The authors have addressed all the requested modifications